# Time-Dependent Seismic Fragility of Typical Concrete Girder Bridges under Chloride-Induced Corrosion

**DOI:** 10.3390/ma15145020

**Published:** 2022-07-19

**Authors:** Xiaoxiao Liu, Wenbin Zhang, Peng Sun, Ming Liu

**Affiliations:** 1School of Civil Engineering and Architecture, Xi’an University of Technology, Xi’an 710048, China; 2Infrastructure Department, Northwestern Polytechnical University, Xi’an 710129, China; zhangwb@nwpu.edu.cn; 3School of Computer Science, Northwestern Polytechnical University, Xi’an 710129, China; sunpeng@nwpu.edu.cn; 4Center for Advancing Materials Performance from the Nanoscale (CAMP-Nano), State Key Laboratory for Mechanical Behavior of Materials, Xi’an Jiaotong University, Xi’an 710049, China

**Keywords:** corrosion, multi-deterioration mechanics, prestress and cracking, system seismic fragility, RC concrete girder bridges

## Abstract

Recent studies highlighted the importance of the combined effects of prestress loss and corrosion deterioration for concrete girder bridge structures when the effect of damage on the performance level is estimated. The multi-deterioration mechanisms connected with chloride erosion include the cross-sectional area loss of longitudinal bars and stirrups, the reduction in the ductility, the decrease in the strength of steels and the strength loss of concrete in RC columns. For the corroded RC columns and corroded elastomeric bridge bearings, analytical models of the material degradation phenomena were employed for performing the probabilistic seismic performance analysis, which could obtain the system seismic fragility of aging bridges by considering the failure functionality of multiple correlated components (e.g., columns, bearings). The combined effects of prestress loss and cracking were also considered when developing time-dependent system seismic fragility functions. Here, a typical multi-span reinforced concrete girder bridge was used as a case study for studying the time-variant seismic performance. The results revealed the importance of the joint effects of the multi-deterioration mechanisms when modeling the time-dependent seismic fragility of aging bridge systems, as well as the significance of considering the combined effects of prestress loss and cracking.

## 1. Introduction

Reinforced concrete (RC) highway bridges experience prolonged exposure to a chloride environment such that these structures suffer from various forms of lifetime degradation, which result in some changes in the material properties. Corrosion in reinforcing steels initiated by chloride ions penetration into the concrete will cause the loss of cross-sectional area of the steels and the cover cracking of RC columns. Some efforts were made to investigate the potential significance of the deterioration of seismic performance across a large number of aged bridges. Choe et al. [1,2] researched the capacity reduction of RC columns in a corroded single-bent bridge due to corrosion and developed a time-variant seismic fragility model of the RC columns; the results revealed the potential importance of the crucial material properties degradation and corrosion parameters. Li et al. [3] performed a series of experiments to evaluate the seismic fragility of corroded RC bridges. Alipur et al. [4] discussed the changes in seismic vulnerability of a group of detailed bridge models, and the effect of the corrosion process on the nonlinear time-dependent parameters was considered during the life-cycle cost. Recent work by Ghosh and Padgett [5] examined the joint effects of corrosion deterioration of the column and steel bearing on the seismic fragility of bridge systems. Li et al. [6] proposed a time-dependent fragility analysis method for bridges under the action of chloride ion corrosion and scour by determining the probability distribution types and statistical characteristics of various environmental and corrosion parameters. To quantify the effect of steel corrosion on seismic fragility estimates, Cui et al. [7] developed an improved time-dependent seismic fragility framework by taking into account the increase in the corrosion rate after concrete cracking and the reduction in the seismic capacity of RC bridge substructures during the service life. Meanwhile, the authors proposed an analytical method based on a backpropagation artificial neural network to provide probabilistic capacity estimates of deteriorating RC substructures. Li et al. [8] proposed a time-dependent seismic fragility assessment framework by considering the correlation of random parameters of aging highway bridges under non-uniform chloride-induced corrosion attacks. Abbasi et al. [9] developed and compared the overall time-dependent system and individual component fragility curves of older and newly designed multi-frame reinforced concrete bridges in California. Dey and Sil [10] proposed performing seismic fragility analysis of corrosion-affected bridges located in the coastal region of India by considering pitting corrosion as a realistic corrosion degradation mechanism. Fu et al. [11] developed a life-cycle fragility assessment method to establish the time-dependent seismic fragility curves of an illustrative cable-stayed bridge at the component and system levels. Fan et al. [12] established a fragility analysis framework for RC bridge structures subjected to the multi-hazard effect of vessel collisions and corrosion. Meanwhile, the modeling and analysis of the time-dependent seismic fragility due to the corrosion effects of RC bridges has received significant attention [13,14,15,16,17,18]. 

However, it should be noted that the above contributions neglect additional mechanisms, such as stirrup corrosion, cover defects and damaged confined concrete. Specifically, the corrosion of stirrups leads to the strain reduction of confined concrete (or the reduction in confinement levels of core concrete) and the reduction of the shear strength in the concrete components (i.e., the bottom and top of bridge columns [19]). Furthermore, the corrosion of steel reinforcements and prestressing strands of the concrete bridges leads to significant prestress losses and cracking, which induces excessive deflections during their service life [20,21]. Meanwhile, expansive forces that originate from the formation of the oxidation product may cause cracking of the concrete cover [22], and the strength of the concrete material is reduced at specific intervals during the life cycle. The deterioration mechanisms of different components of RC bridges depend on their physical conditions and environmental exposure scenarios. For multi-span continuous (MSC) bridges and multi-span simply supported (MSSS) girder bridges, the different forms of degradation due to corrosion can be found in the individual components, and the multiple types of deterioration among different components (i.e., column, bearing) of RC bridges should be considered as dependent. Meanwhile, the corrosion degradation of RC components of typical bridges mainly gives rise to the cross-sectional area loss of embedded steels, including the longitudinal bars and hoops. This type of degradation can also lead to stress reduction of unconfined concrete and the ductility decrease of confined concrete, along with cracking of the cover. These multi-deterioration mechanisms are common for bridge components, such as the decks and columns of RC bridges. In addition, the diameter loss of steel bearing anchor bolts and the strength of elastomeric bearing dowel bars can cause the deterioration of the bearing system. Some studies [5,23] revealed that the corrosion of the reinforcement steel in RC columns and the deterioration of the bridge bearings could be found in MSC steel girder bridges and MSSS concrete girder bridges in Central and Southeastern United States, which are seismically vulnerable regions. According to China’s Ministry of Transportation, the aging and deterioration phenomenon of MSC concrete girder bridges are commonly found in seismic zones, such as northern China, because deicing salts are extensively used on highway bridges [24]. The existing contributions [6,7,8,9,10,11,12,13,14,15,16,17,18,19,20,21] also indicated that RC bridges subjected to the mechanisms of corrosion deterioration might be more vulnerable to earthquake hazards. Moreover, since previous studies [25,26] demonstrated that chloride from deicing salts can cause more degradation of bridges than that in a marine environment, the impacts of corrosion degradation due to chloride-laden deicing salts should be researched in detail.

Although the above works focused on the evaluation of the corrosion effect on the seismic vulnerability of RC bridge components/systems, the joint impacts of the multi-deterioration mechanisms among multiple bridge components, which have a significant impact on the seismic performance, were rarely considered in recent studies. Meanwhile, the corrosion of critical components, including the corrosion of RC columns and the aging of elastomeric bridge bearings, as well as the combined effects of prestress loss and cracking, also have a negative effect on seismic performance. Therefore, bridge columns are assumed to be subjected to salt mist formed from chloride-laden water [27], and the elastomeric bridge bearings were considered to be affected by chloride ions and thermal oxidation [28,29,30] in this study. For this purpose, the joint effects of multi-deterioration mechanisms on system seismic fragility for RC bridges were researched by considering the degradation correlations between multiple components. Then, dynamic behaviors of bridge systems under multiple related deterioration mechanisms could be calculated to support the development of time-variant seismic fragility curves. Subsequently, a probabilistic approach was proposed to evaluate the time-dependent seismic fragility, and the effects of multi-deterioration mechanisms on seismic vulnerability could be researched in detail by including the joint impact of columns and bearing corrosion deterioration. The multi-deterioration mechanisms, which are connected with chloride erosion, include the cross-sectional area loss of longitudinal bars and stirrups, the reduction in the ductility, the decrease in the strength of steels and the strength loss of concrete in RC columns because the cover cracking exists and the failure of the bearings system of bridges will appear due to the corrosion of bearing pads and dowel bars. The impact of multi-deterioration were applied to a typical MSC concrete girder bridge in the system fragility assessment framework of aged bridges. Meanwhile, the combined effects of the prestress loss and cracking on the seismic fragility were found in detail. For the corroded RC columns and corroded elastomeric bridge bearing, analytical models of material degradation phenomena were introduced into the probabilistic seismic performance evaluation, which obtained the system fragility of bridges by considering the failure functionality of multiple correlated critical components (e.g., columns, bearing). A three-dimensional nonlinear dynamic model of the MSC bridges, which considered its aging components as multi-deterioration mechanisms, was performed to evaluate the changes in the structural capacity and was also employed to obtain the seismic response of critical components and systems. Subsequently, time-dependent seismic fragility curves were analyzed to generate a comparison of the seismic performance at different time points during the service life when the parameter properties’ variation, ground motions and deterioration parameters were considered simultaneously. The results showed that the multi-deterioration of the multiple correlated components had a joint effect on the seismic performance of the aged bridges subjected to seismic hazards, as well as the combined effects of prestress loss and cracking.

## 2. MSC Concrete Girder Bridge Geometry

To verify the proposed methodology for developing time-dependent fragility curves and offer insights on the effects of the multi-deterioration mechanism of multiple related components on seismic response and vulnerability, a sample MSC concrete girder bridge was considered as a case study in this research. Nearly 67.8% of all bridges in Central and Southeastern United States and approximately 74% in China are of this bridge type [24]. The MSC concrete girder bridge type was used for the case study since an overwhelming majority of 74% of bridges found in seismic zones are of this type and these bridges are seismically vulnerable due to inadequate detailing of the components. Previous studies on classes of bridges by Nielson [31] indicated that the vulnerability of multiple components of this type of bridge is necessary when considering aging and deterioration. This vulnerability phenomenon can be attributed to inadequately seat widths, bolted elastomeric pad bearings and insufficient transverse reinforcement, which inhibits the shear resistance and ductile capacity in RC columns. Consequently, higher seismic demands on the bridge system due to the above reasons can be generated during earthquake events. 

The typical MSC concrete girder bridge identified by Nielson [31] is introduced in this study because of the similar bridge configuration in China [24]. As shown in Figure 1, this bridge has three spans with lengths of 11.9, 22.3 and 11.9 m, which give an overall bridge length of 46.1 m. The decks of width 15.01 m were constructed of eight AASHTO concrete bridge girders. Among these girders, type I was for the end span and type III was used for the middle span. A concrete parapet between the girders and deck, which can reduce the effects of live load, was used to concatenate the MSC bridge [31]. Two pile bent abutments and two three-column bents support a three-deck. The circular columns have a cross-sectional area of 641.2 mm^2^. This type of column can be reinforced with 12-#29 longitudinal bars and #13 transverse stirrups spaced at 305 mm. The fixed and expanded bearings for this bridge class are elastomeric pads with two steel dowels, as characterized by Neilson [31].

## 3. Multi-Degradation Mechanisms of RC Columns Due to Corrosion

All equations used in Section 2 can be found in Table 1. 

### 3.1. The Diffusion Process and Corrosion Initiation Time

Chloride-induced corrosion is considered to be one of the major causes of the aging of reinforced concrete bridges. Corrosion of reinforcing steel in an RC column is initiated by the ingress of chloride ions through the concrete cover to the bar’s surface. The main source of chloride ions is considered to be the deicing salts used in winter. The diffusion process of chloride ions that result in corrosion can be described by Fick’s second law [25]. The formulation can be seen as Equation (1) in Table 2, where C is the chloride ion concentration, D is the chloride diffusion coefficient of concrete, x is the depth of the concrete cover and t is the time step in years. We assumed that the initial condition (initial chloride ion content), boundary condition (surface chloride content) and material property (chloride diffusion coefficient) were equal to zero, the mean-invariant $C_{s}$ and the mean-invariant D, respectively. The closed solution of the chloride ion content is shown as Equation (2) [32], where erf(⋅) is the Gaussian error function. The corrosion parameters involved in the diffusion process were assumed to be random variables whose distribution type, mean u and coefficient of variation COV are listed in Table 2 [5,19]. 

The diffusion process of chloride ions can be described in probabilistic terms by using the Monte Carlo method. It was assumed that the depth of concrete cover x was 40 mm and the service life t was 50 years; the resulting probability density functions (PDFs) of chloride ion concentrations that were obtained using a Monte Carlo simulation with a sample size of 50,000 are shown in Figure 2. With reference to Fick’s second law of diffusion and Equation (2), the corrosion initiation time $T_{I}$ was evaluated using Equation (3) [5], where $T_{I}$ is the corrosion initiation time in years, $C_{S}$ is the chloride concentration of concrete surface and $C_{C}$ is the critical chloride concentration that can dissolve the protective passive film around the reinforcement steels. Therefore, a Monte Carlo simulation was performed with a sample size of 50,000 for random variables to evaluate the probability distribute of the corrosion initiation time $T_{I}$, as shown in Figure 3. A lognormal distribution with a mean μ = 1.0374 years and standard deviation σ = 0.2829 was found to be a good estimation of the simulated distribution for the corrosion initiation time $T_{I}$. This distribution had an influence on the probabilistic modeling of the steel corrosion in the bridge RC columns. 

### 3.2. Reduction in the Cross-Sectional Area of Longitudinal Bars and Stirrups

The corrosion of rebars in the existing RC column was initiated by reducing the cross-section. However, some recent studies [33,34] showed that the corrosion of stirrups, which causes a reduction in confinement behavior, is more severe than that of the longitudinal bars in an RC column. Furthermore, the corrosion can degrade the shear resistance of the column by potentially shifting the failure modes from a ductile failure mode to a shear failure mode. Therefore, the effect of both the loss of the cross-sectional area and the ductile degradation of stirrups on the seismic behavior of bridge systems should be considered in this study. Corrosion from deicing salts leads to a reduction in the effective area of both the longitudinal and transverse rebars. First, the corrosion penetration R is introduced as Equation (4) [19], where $r_{corr}$ is the corrosion rate and represents the implicit function of the time-dependent area due to corrosion, while $t-T_{I}$ is the corrosion propagation time. The time-variant loss of longitudinal reinforcement and stirrup diameter in the BLG model can be evaluated using Equation (5) and Equation (6), respectively [19]. Combining Equation (4), Equation (5) and Equation (6), $\Phi \left(t\right)$ and $\phi \left(t\right)$ can be expressed as Equation (7) and Equation (8), where $\Phi \left(0\right)$ and $\phi \left(0\right)$ are the initial diameters of the longitudinal bar and stirrup, respectively; $\Phi_{i}$ and $\phi_{i}$ are the diameters of a corroding longitudinal bar and stirrup at time t=i, respectively; $\Phi \left(t\right)$ and $\phi \left(t\right)$ are the diameters of a longitudinal bar and stirrup at the end of $t-T_{I}$ years. The corrosion penetration index of rebars η is introduced as Equations (9) and (10) [35]. 

Then, the time-dependent area of a corroding longitudinal bar can be expressed as Equation (11), where $A_{S}\left(0\right)$ is the initial area of longitudinal reinforcement and $\eta_{S}\left(t\right)$ is the loss rate of a longitudinal bar. The time-dependent area of a corroding stirrup can be expressed in Equation (12), where $a_{s}\left(0\right)$ is the initial area of a stirrup and $\lambda_{k}\left(t\right)$ is the loss rate of a stirrup. It is acknowledged that the corrosion rate is considered a constant on average and it is assumed to be lognormally distributed, as listed in Table 1. Moreover, the diameters of the longitudinal rebar and stirrup are assumed to be lognormally distributed, whose mean u and coefficient of variation COV are listed in Table 3. On the basis of the above assumptions, the reduction in the diameter of the rebar due to corrosion could be evaluated for a lifetime of 50 years ($\Phi \left(0\right)$ = 28.58 mm and $\phi \left(0\right)$ = 12.70 mm). The time-variant mean of the longitudinal bar and the stirrups, as well as the corresponding standard deviations, are shown in Figure 4. The scattering region around the mean represents the stochastic time-dependent loss of both the longitudinal bars’ and the stirrups’ cross-sectional areas because of the effect of uncertainties in degradation parameters. This figure illustrates that the effects of stirrup corrosion were more significant than that of longitudinal reinforcement corrosion because the smaller diameter of a transverse bar led to higher levels of corrosion of the stirrup. Thus, the area loss of steel corrosion in RC columns could be modeled as the reduction in the cross-sectional area of both longitudinal reinforcing and stirrups in the BLG model with the fiber section. 

### 3.3. Reduction in Strength and Ductility of Corroded Longitudinal Bars and Stirrups

The corrosion of stirrups is more serious than that of longitudinal reinforcement in RC columns and it leads to a reduction in confinement behavior. The corrosion can degrade the shear-resistant capacity of RC columns by potentially changing the ductile failure to brittle failure or even shear failure [34]. According to some studies [36,37,38], the reduction in both the strength and the ductility in longitudinal reinforcement can be considered an explicit function of the cross-sectional loss. The cross-sectional loss rate of the longitudinal bar is considered a function of the corrosion penetration index η in Equation (13) [19]. Therefore, the time-dependent ultimate strain of the longitudinal reinforcement in Steel 01 material can be expressed as Equation (14) [19], where $\varepsilon_{su}\left(0\right)$ is the initial nominal value of the ultimate strain of the longitudinal rebar. The time-variant ultimate strain is shown in Figure 5.

Furthermore, the residual yield strength of the longitudinal rebar depends on the cross-sectional loss rate $\eta_{S}\left(t\right)$, which is expressed as Equation (15) [39], where $f_{sy}\left(0\right)$ is the initial nominal yield strength of the longitudinal bar. Similarly, the time-dependent yielding strength of the hooping in the BLG model depends on the cross-sectional loss rate $\lambda_{k}\left(t\right)$. The yielding strength is expressed as Equation (16) [39], where $f_{yh}\left(0\right)$ is the initial yielding strength of the stirrups. The yielding strengths of both the longitudinal bar and stirrup were assumed to be lognormally distributed, whose mean u and the coefficient of variation COV are listed in Table 1. The time-dependent yielding strengths $f_{sy}\left(t\right)$ and $f_{yh}\left(t\right)$ can be respectively evaluated using the cross-sectional loss rate $\eta_{S}$ and $\lambda_{k}$, which depend on the reduction in the diameter in both the corroding longitudinal rebar and hooping. 

### 3.4. Reduction in Strength and Ductility in Corroded Concrete 

For the corrosion initiation phase, the confinement behavior of the concrete can be enhanced due to the expansion of the corrosion products. However, the propagation of longitudinal cracks and the cracking of concrete cover inversely result in a reduction in the confinement with the accumulation of rust products. According to the unconfined and confined stress–strain rules (i.e., BLG model, Figure 6), the residual strength of the unconfined concrete can be evaluated using Equation (17) [40], where K is a coefficient related to the rebar roughness and the diameter (a value K = 0.1 is used for medium-diameter ribbed rebars), $\varepsilon_{c}\left(0\right)$ is the initial strain of the unconfined concrete at the peak compressive stress $f_{pc}(0)$ and $\varepsilon_{t}\left(t\right)$ is the average tensile strain in the cracked concrete at time t. The average tensile strain can be evaluated using Equation (18) [40], where $n_{bars}$ is the number of longitudinal rebars, $r_{i}$ is the width of the cross-sectional area in a pristine state and w is the crack width for each longitudinal rebar. The relationship between the crack width and cross-sectional loss of the longitudinal steel bar can be expressed as Equation (19) [41], where $k_{w}$ = 0.0575 mm^−1^, $\Delta A_{s}$ is the cross-sectional loss of a longitudinal steel bar for cracking propagation and $\Delta A_{s0}$ is the cross-sectional loss of the longitudinal reinforcement for cracking initiation. Then, one obtains the following expressions: $\Delta A_{s}$ = $\eta_{S}\left(t\right)A_{S}\left(0\right)$ and $\Delta A_{s0}$ = $\eta_{S}\left(0\right)A_{S}\left(0\right)$. The cross-sectional loss rate of the longitudinal rebar for the cracking initiation $\eta_{S}\left(0\right)$ can be evaluated using Equation (20) [40], where $\phi \left(0\right)$ is the initial diameter of the longitudinal rebar, x is the depth of the concrete cover and α is the pit concentration factor. For uniform corrosion, one has α = 2, and for the localized corrosion, one obtains 4 < α < 8. The initial compressive strength of unconfined concrete is assumed to be lognormally distributed, whose mean u and the coefficient of variation COV are listed in Table 1. The residual strength $f_{pc}\left(t\right)$ can be evaluated using the cross-sectional losses $\Delta A_{s}$ and $\Delta A_{s0}$, which are dependent on the swedged steel bars, as shown in Figure 7. 

Since a failure criterion for confined concrete is not provided in the BLG model, the ultimate compressive strain proposed by Scott et al. [42] was assigned to the concrete core fibers. The time-dependent ultimate compressive strain of the confined concrete related to the first stirrup facture can be estimated with the failure criterion. Furthermore, the strain is expressed as Equation (21), where $\rho_{s}(t)$ is the volume–stirrup ratio at time t and $f_{yh0}$ is the initial yield strength of a transverse stirrup. The ultimate compressive strain of the confined concrete $\varepsilon_{cu}(t)$ can be calculated using the stirrup ratio $\rho_{s}(t)$, which has a direct dependency on the percentage loss of the total swedged reinforcement in the cross-section of an RC column, as shown in Figure 8. The result indicated that the ultimate compressive strain of the confined concrete is also time-dependent because the progressive reduction of stirrup area was closer to the concrete surface, which can be more susceptible to corrosion. Therefore, the material degradation of the RC column was properly simulated by using the BLG model and Steel 01.

## 4. Degradation of Elastomeric Bridge Bearings Due to Corrosion and Thermal Oxidation

All equations for Section 3 can be found in Table 4.

Elastomeric bridge bearings are widely used in concrete girders, where they allow for force transmission from the superstructure to the substructure. These types of bearing systems consist of an elastomeric nature rubber (NR) pad and steel dowels, but they often form “walk-out” bearings during seismic events due to the effects of aging and deterioration [43]. Corrosion of these assemblies, caused by chloride-laden water from deicing salts, may potentially result in a larger deformation of the bearing systems under seismic loading. For the steel dowels of the elastomeric bridge bearings, corrosion deterioration can lead to a reduction in both the cross-section and the shear strength. Moreover, the elastomeric NR pads suffer an increase in both the shear modulus and shear stiffness due to corrosion and thermal oxidation [28,29,30].

Additionally, each member of this type of bearing system makes a contribution to the transfer of forces. For example, an elastomeric NR pad transfers a lateral load by developing a frictional force, while steel dowels provide resistance via the action of the beam. The assemblies of the elastomeric bridge bearings consist of the fixed and expansion types, which depend on the dimensions of the slot in the NR pad [31], as shown in Figure 9. The sliding behavior of the elastomeric NR pad and the yield of steel dowels in pristine bridge bearings can be modeled by using the proper composite action. 

As above-mentioned, aging and thermal oxidation may lead to an increase in the shear modulus in the elastomeric NR pads, where these deterioration mechanisms can be modeled with a changed value of the constitutive parameter in Steel 01 materials. Normally, the value of the shear modulus in the elastomeric bridge bearings recommended by AASHTO [44] is assumed to be a mean constant in the analytical modeling of bridges. However, a battery of accelerated exposure tests proposed by Itoh et al. [28] illustrated that the shear modulus of the NR pad is not a constant and can be changed over time due to thermal oxidation. The derivation process of the time-dependent shear modulus for an NR pad can be expressed as follows. 

First, an aging model of NR bearings proposed by Itoh and Gu [29,30] showed the relation between the variation of strain energy and the aging time under accelerated exposure test conditions (i.e., the simulated thermal oxidation). This aging model can be provided by Equation (22), where $B_{s}/B_{0}$ represents the relative strain energy versus its initial state at the NR block surface, $k_{s}$ is the strain-dependent coefficient for the strain energy and $t_{r}$ is the aging time at the test temperature in hours. Here, 60 °C was taken as the reference temperature of the NR, and the empirical formula for $k_{s}$ was calculated using Equation (23), where $a_{1}$ = 0.54, $a_{2}$ = −4.19, $a_{3}$ = 8.16 and $a_{4}$ = 9.59 at 60 °C, and ξ is the nominal strain. 

Second, a relationship between the strain energy and the shear modulus can be expressed using a one-parameter neo-Hoolean material model [45], as shown in Equation (24), where B is the strain energy; G is the shear modulus; and $\lambda_{1}$, $\lambda_{2}$ and $\lambda_{3}$ are the stretches due to the uniaxial tension. If this material is incompressible, then $\lambda_{1}^{2}\lambda_{2}^{2}\lambda_{3}^{2}=1$. Combining Equations (22) and (23), a relative change in the shear modulus of the NR bearing due to thermal oxidation can be calculated using Equation (25), where $G_{s}/G_{0}$ is the relative shear modulus versus its initial state for the NR bearing at time $t=t_{r}$. Thus, the normalized shear modulus variation $\Delta G_{s}$ can be obtained using Equation (26). Finally, the shear modulus variation in the outer region from the NR bearing’s surface to the critical depth h should be expressed by using an equation. The relationship between the critical depth and the temperature can be obtained using Equation (27) [29], where $T^{*}$ is the absolute temperature and α and β are the coefficients determined by the thermal oxidation test. It is assumed that the shear modulus variation $G(t)/G_{0}$ might be a function of the position m. The boundary conditions can be derived using Equations (28) to (30), where G(t) and $G_{0}$ are the shear modulus at time t and the initial state, respectively; $\Delta G_{s}$ is the normalized shear modulus variation; and l is the width of the NR bearing. The normalized shear modulus $G(t)/G_{0}$ can be also assumed to be a square relation of the position m, and the function is given as Equation (31). Combining the boundary conditions, the normalized shear modulus $G(t)/G_{0}$ can be written as Equations (32) and (33). The time-dependent shear modulus in a bearing NR pad can be calculated using Equation (34), where G(t) is the shear modulus at real time t; τ is coefficient correlated with the position m, the critical depth h and the width l of the bridge NR bearing. Since thermal oxidation is commonly assumed to be a first-order chemical reaction for NR materials, the relationship between the deterioration time under the service condition and the time in the accelerated exposure tests can be expressed as Equation (35) [29], where $t_{r}$ is the test time, t is the aging time, $T_{r}$ is the absolute temperature in the thermal oxidation test, T is the absolute temperature under the service condition, R is the gaseous constant (8.314 J/mol K) and E is the activation energy of the rubber (94,900 J/mol). Consequently, the normalized shear modulus variation $\Delta G_{s}$ can be written as Equation (36). 

The initial shear modulus of the elastomeric bearing pad is assumed to be uniformly distributed, whose upper bound a and lower bound b are listed in Table 1, and the time-dependent shear modulus G(t) is plotted in Figure 10. However, the time-variant shear stiffness of the NR pad is considered a critical element in the analytical modeling of elastomeric bearing pads, and the aging model of the shear stiffness can be expressed as Equation (37), where G(t) is the shear modulus of the NR pad at the real time t, $A_{p}$ is the area of the elastomeric bearing *NR* pad and $S_{p}$ is the thickness of the bearing pad. For the assemblies of the fixed and the expansion steel dowels, corrosion deterioration may lead to a loss of the cross-sectional area of this component type. Then, this deterioration effect is modeled by considering the variation of parameters in hysteretic material, which includes a reduction in the yield strength and ultimate lateral strength. The time-dependent yield strength and the ultimate lateral strength can be calculated using Equations (38) and (39) [46], where $f_{y}$ and $f_{u}$ are the tensile strength and the ultimate shear strength of the steel dowels, respectively, and $A_{d}(t)$ is the time-dependent cross-sectional area of the steel dowels. The initial area of the steel dowels and dowel tensile strength, as well as the lateral strength of the bearing dowel, are assumed to be uniformly distributed and whose upper bound a and lower bound b are listed in Table 1, respectively. Thus, the yield strength $F_{y}(t)$ and the ultimate lateral strength $F_{u}(t)$ can be evaluated using the cross-sectional loss of dowel bars $A_{d}(t)$, as shown in Figure 11. The results indicated that a reduction in the dowels’ area has a significant impact on the strength over time due to corrosion.

## 5. Impact of Corrosion on the Seismic Response of Bridge Components

All equations for Section 4 can be found in Table 5.

To analyze the impact of corrosion multi-deterioration mechanisms on the seismic fragility of the MSC concrete girder bridge, the seismic responses of the MSC bridge components, which considers the effect of the time-dependent aging, should be presented using dynamic simulations. For the geometry of this bridge type considered in a pristine state, the first two fundamental modes along longitudinal and transverse directions were 0.49 s and 0.37 s, respectively. Moreover, the foundations for the MSC bridge were assumed to be on medium hard soil represented by site class II (shear wave velocity between 500 m/s and 250 m/s), and the target response spectrum of 0.53 s was calculated by considering the site soil conditions. Then, twenty real earthquake records, whose response spectra had the greatest similarity to the target response spectrum, were selected to capture the uncertainty of ground motions. Thus, the dynamic responses of the bridge were illustrated through nonlinear time history analysis with the above twenty seismic inputs. The impact of corrosion multi-deterioration mechanisms on the evolving dynamic response of the bridge is presented using probabilistic seismic demand models (PSDMs) of aging components (Equation (42)). The influence of multi-deterioration mechanisms of a single component and the joint effects of both column and bearing on the seismic demands are compared in the following sections.

### 5.1. The Impact of Deterioration on the Seismic Demand of RC Columns 

As elaborated on earlier, the aging of RC columns is modeled via a relationship between stress and strain, which includes multiple deterioration mechanisms. There is a great shift in the demands of the aging RC columns due to the loss of both the steel area (i.e., longitudinal bars, stirrups) and concrete cover when the bridge experiences seismic loading. Then, the seismic demand placed on the column can be expressed using Equation (40), where $u_{\theta }$ is the curvature ductility demand ratio, $\theta_{m}$ is the maximum curvature demand of a column under seismic loading and $\theta_{y}$ is the yield curvature in the column. The peak curvature ductility demands at 0 years and 50 years are shown in Figure 12. It can be seen that if only column multi-degradation was considered, it had a great impact on the column demands, which were significantly increased with aging from 0 to 50 years. Additionally, when the joint effects of both the column and bearing multi-deterioration were taken into account, the column demands at a specific time can be higher than when the multi-deterioration of an individual column component is considered. It is worth noting that the only bearing multi-deterioration had a minimal impact on the column demands. The results indicated the significance of considering the multi-deterioration mechanisms of the multiple components.

### 5.2. The Impact of Deterioration on the Seismic Demand of Bridge Bearings

Similarly, Figure 13 shows the influences of the multi-deterioration of the multiple components and a single component on the seismic response of the fixed bearing along the longitudinal direction. When only the multi-degradation of the fixed bearing was considered, there was a significant influence on the bearing deformation, which constantly increased from 0 to 50 years. If only column multi-degradation was considered, there was a small impact on the bearing deformation. Both the joint considerations of the column and bearing degradations did not have a higher effect on the bearing demand along the longitudinal direction than when only bearing deterioration is considered. However, Figure 14 shows the dependency between the multi-deterioration mechanisms of the column and expansion bearings in the transverse directions.

If only column multi-deterioration is considered, the demands on the expansion transverse bearing at 50 years were lower than the demands on the pristine expansion bearing due to the domination of the corroded RC column. When the multi-deterioration of the expansion transverse bearing was considered, the demands on the expansion bearing at 50 years were higher than that of the pristine demands. The reason for this was that the increase in strength reduction of steel dowels was higher than the increase in the shear stiffness of the NR pad induced by thermal oxidation. The multi-deteriorations of both the column and the expansion transverse bearing had a greater impact on the bearing deformation than when only column deterioration was considered. However, the deterioration mechanisms of multiple components had a lower influence on the bearing deformation than when only bearing degradation was taken into account. The reason for this was that the individual consideration of the corrosion degradation of the expansion bearing system had a greater role on the seismic responses after 50 years of exposure to deicing salts.

## 6. Impact of Corrosion on the Seismic Fragility of an Aged Bridge System

All equations for Section 5 can be found in Table 5.

The impact of multi-deterioration mechanisms on the seismic performance of aged bridge components can be evaluated by developing time-dependent bridge fragility curves at a system level. Moreover, the increase in the probability of a damaged state exceedance with continuous degradation of the bridge along its service life can be quantified using such fragility curves. A methodology for the development of time-dependent fragility curves is presented in the following section using a typical case study of an aged MSC concrete girder bridge. The uncertainties in bridge modeling attributes, ground motion and deterioration parameters were considered in probabilistic terms for this case. 

### 6.1. Time-Dependent Probabilistic Seismic Demand Models (PSDMs)

Time-dependent fragility curves represent the probability of damage exceedance of structures under earthquake excitation at different points in time throughout the service life. Such time-evolving fragility curves show the impact of multi-deterioration mechanisms on the seismic performance of aging structures at the system level. The generalized time-dependent seismic fragility function can be expressed as Equation (41), where $P_{f}\left(t\right)$ is the probability of damage state exceedance of a specific MSC bridge at an aging time t. $D\left(t\right)$ and $C\left(t\right)$ are the seismic demand and the capacity of an aged bridge at time t, respectively. IM is the intensity of the ground motion. Before developing the time-dependent seismic fragility curves, the relationship between the time-varying seismic demand and capacity should be established by using probabilistic seismic demand models (PSDMs). Such PSDMs for bridge critical components are constructed using nonlinear time history analysis to capture the impact of multi-deterioration mechanisms on the dynamic responses. Thus, to consider the uncertainties in the ground motion and structural parameters, a total of 20 real ground motions from PEER were used in the analysis and an equal number of random bridge samples in pristine states were generated through Latin hypercube sampling. The peak ground acceleration (PGA) of the 20 random samples was modulated ranging from 0.095 g to 1.05 g. In addition, to generate 20 random aging bridge samples, the uncertainties in the corrosion parameters that affect the multi-deterioration mechanisms of the column and bearing components were also taken into account in the finite element modeling.

To develop probabilistic seismic demand models for bridge components at different points in time (e.g., 0, 25 and 50 years), the 20 random bridge samples for each time instant at a specific intensity level of the ground motion were performed using nonlinear time history analysis. Then, probabilistic seismic demand models, which reflect the relationship between the peak demands of aged bridge components and ground motion intensity, were developed using linear regression. The demands considered in this research included RC columns and fixed and expansion elastomeric bearings. Consequently, the time-dependent probabilistic seismic demand models can be expressed as Equation (42) [31], where ${\hat{D}}_{i}\left(t\right)$ is the time-dependent median value of the seismic demand for bridge component i, $a_{i}\left(t\right)$ and $b_{i}\left(t\right)$ are the linear regression parameters, and IM is the intensity of ground motion. A logarithmic seismic demand for the specific component $\ln D_{i}\left(t\right)$ was assumed to be normally distributed, and then $\ln D_{i}\left(t\right)$ and $\beta_{Dm,i}\left(t\right)$ were the median value and dispersion, respectively. $\beta_{Dm,i}\left(t\right)$ can be expressed as Equation (43), where $d_{m}\left(t\right)$ is the mth peak seismic demand for bridge component i and $IM_{m}$ is the mth PGA. Then, the time-evolving PSDMs can also be expressed by using Equation (44), where $\lambda_{i}\left(t\right)$ is the natural logarithm of the seismic demand related to the median value of PGA and $\lambda_{i}\left(t\right)=\left(\ln d_{i}\left(t\right)-\ln a_{i}\left(t\right)\right)/b_{i}\left(t\right)$; $\xi_{i}\left(t\right)$ is the lognormal standard deviation and $\xi_{i}\left(t\right)=\beta_{Dm,i}\left(t\right)/b_{i}\left(t\right)$.

### 6.2. Time-Dependent Seismic Fragility of an Aged Bridge System

In addition to probabilistic seismic demand models, the structural capacities of different bridge components should also be estimated during time-dependent seismic fragility analysis. The limit state capacities of different components considered in this study are lognormal and presented in Table 6 [31]. When the seismic demands and the limit state capacities for different bridge components are assumed to be lognormally distributed, the time-dependent seismic fragility at the component level can be obtained using Equation (45), where $\gamma_{i}\left(t\right)$ and $\zeta_{i}\left(t\right)$ are median values (in units of g PGA) and logarithmic standard deviations of the ith component fragility, respectively. $\gamma_{i}\left(t\right)$ and $\zeta_{i}\left(t\right)$ can be expressed as Equations (46) and (47), where $\overset{\wedge }{C_{i}\left(t\right)}$ and $\beta_{C,i}\left(t\right)$ are the median and dispersion of the ith component capacity, respectively, and $\beta_{Dm,i}\left(t\right)$ is the dispersion of the ith component demand.

The assessment of bridge system vulnerability is performed by assuming the bridge as a series system, as presented by Nielson [47,48]. The demands of the bridge components under seismic loading are considered dependent and then the correlation coefficient between the peak responses can be estimated by constructing a joint probability density function (JPDF) for component demands. The generalized formula for the aged bridge system fragility can be derived using joint probabilistic seismic demand models (JPSDMs) [47,48]. The JPSDMs can be written as Equation (48), where $\gamma_{sys}\left(t\right)$ and $\zeta_{sys}\left(t\right)$ are the median values (in units of g PGA) and logarithmic standard deviations of the system fragility at different points in time, respectively. Solutions to Equation (48) can be directly calculated by using Monte Carlo analysis. 

### 6.3. Time-Dependent Seismic Fragilities of Aged Bridge System

The joint impacts of the multi-deterioration of multiple components on system vulnerabilities are quantified by developing time-dependent fragility curves of the overall bridge. Figure 15 shows the aging bridge system fragility curves at three different points in time (e.g., 0, 25 and 50 years) for moderate and complete states. It can be clearly seen that the seismic fragility curves at the system level increased steadily with age from 0 to 50 years. However, the seismic fragility of the individual components tended to show a reduction in vulnerability when the bridge component continued to be corroded due to the individual consideration of multi-deterioration for a single component. Furthermore, the seismic fragility of the bridge expansion bearing along the transverse direction can decrease at the initial time (e.g., 0, 15 and 25 years) due to the added stiffness of the bearing NR pads induced by thermal oxidation. The results indicated the importance of considering the joint effects of multi-deterioration mechanisms for multiple bridge components. Furthermore, it was found that the multi-deterioration of the aging MSC concrete girder bridge had a potentially negative influence on the overall seismic fragility at the system level.

### 6.4. Time-Dependent Seismic Fragilities Considering Prestress Losses and Cracking

A reference model for the corrosion-induced presetress loss was developed [20,21] by considering the effects of concrete cracking. Corrosion-induced prestress loss can be modeled as the difference between the effective prestress in an uncorroded strand and that in the corroded strand [49]. Here, it was suggested that the strain compatibility and force equilibrium equations in concrete bridge-girders could also be used to evaluate the effective prestress in the corroded strand. If the pre-stressing force in the pre-tensioned concrete bridge-girders is released, then the strand pre-stress would transfer to the concrete via the bonding stress at the stand-concrete interface. Then, the effective prestress in the corroded strand can be calculated using Equation (49) [49], where $T_{p}$ represents the tension force of the corroded strand and $A_{p}\left(\eta \right)$ denotes the residual cross-sectional area of the corroded strand. 

During the corrosion process, RC components of bridges also suffer from the prestress and the expansive pressure. When the tensile stress induced by the expansive pressure exceeds the concrete tensile strength, the concrete is considered to be cracking. The concrete cover can contain a cracked inner region and an uncracked outer region. Here, the outer wires can be considered to be first corroded when the strand suffers from corrosion. The corrosion loss of a strand can be expressed as Equation (50) [50], where *R*_0_ and $R_{\rho }$ are the radiuses of wire before and after corrosion, respectively, and $A_{p}$ is the strand cross-sectional area. Meanwhile, it is assumed that the smeared cracks in the cracked region are distributed uniformly, and then a reduction factor can be used to reflect the residual tangential stiffness in the cracked concrete. Consequently, by combining stress equilibrium equations with the strain compatibilities, the crack width on the concrete surface can be written as Equation (51) [50], where *R_t_* is the radius of wire with corrosion products; $R_{c}=R_{0}+C$, where *C* is the concrete cover; $\nu_{c}=\sqrt{\nu_{1}\nu_{2}}$, where $\nu_{1}$ and $\nu_{2}$ are the Poisson ratios of concrete in the radial and tangential directions, respectively; *a* is a reduction factor; $f_{t}$ is the concrete tensile strength under the biaxial stress state; and *E_c_* is the elastic modulus of concrete.

Subsequently, the combined effects of the prestress losses and cracking on the seismic fragilities of bridge systems can be quantified by using Equation (48), which also includes the effects of the multi-deterioration of multiple components. Figure 16 shows the aging bridge system fragility curves at three different points in time (e.g., 0, 25 and 50 years) for moderate and complete states. It can be derived from Figure 15 and Figure 16 that the system fragility when considering the combined effects of prestress losses and cracking was more conservative than that without considering these effects. The results indicated that the combined effects of the prestress losses and cracking should not be neglected when performing the seismic fragility of aging bridge systems. 

Finally, it is stressed that other time-variant approaches, such as the empirical experiment and trial-and-error [51,52,53], for the corrosion process of aging bridge systems are very expensive and very unrealistic when it comes to obtaining a series of multi-deterioration mechanisms among multiple components of aging bridge systems. Moreover, the empirical experiment and trial-and-error method are very resource-consuming when dealing with the time-dependent system fragility of such complex aging bridges when undergoing the earthquake excitation test. Therefore, when we would like to efficiently develop the overall time-variant deterioration process of aging bridge systems, the finite element modeling combined with the theoretical modeling of the corrosion process is the best choice for performing the time-dependent overall seismic fragility curves of aging bridge systems. The applicability of different approaches can be summarized in Table 7.

## 7. Conclusions

This paper provides a probabilistic method for identifying the time-dependent fragility of an aging bridge system under earthquake events by considering the impacts of multi-deterioration of multiple bridge components, as well as the combined effects of the prestress losses and cracking. A typical MSC concrete girder bridge is used to evaluate the seismic performance by considering the uncertainty models in structural material and corrosion parameters. Chlorine corrosion from deicing salts, which are widely utilized across northern China, can be considered the cause of the degradation and aging of bridge systems. Here, the multi-deterioration mechanisms include the corrosion deterioration of an RC column due to loss of both the steel area (i.e., longitudinal bars, stirrups) and concrete covers, and the deterioration of elastomeric bearing assemblies due to the reduction in both the shear stiffness of bearing NR pads and steel dowel area. The multi-deterioration mechanisms of RC components of the bridge systems affect the lateral force resistance and seismic responses. In addition, the models for corrosion-induced prestress loss and cracking are introduced to develop time-dependent seismic fragilities of bridge systems. The overall seismic demands found using PSDMs showed a steady increase as the growing service life when joint effects of multi-deterioration of bridge multiple components (e.g., RC columns and fixed bearings) were considered. However, there was a decrease in the demand (e.g., bearing deformation) on other multiple components (e.g., RC columns and expansion transverse bearings). 

Time-variant PSDMs and the components’ capacities were constructed for calculating the seismic fragility of the MSC concrete bridges, which considered the variability in attributes of bridges, ground motion and degraded parameters. The seismic fragility curves at the system level demonstrated a steady increase over time due to corrosion aging. However, individual components, such as expansion bearings in the transverse direction, revealed a decreased vulnerability at the initial time due to the added bearing pad shear stiffness. Subsequently, the dominant effect of the loss of steel dowel areas led to an increase in vulnerability. Moreover, the system seismic fragility considering the combined effects of prestress losses and cracking was more conservative. That meant the effects of both prestress losses and cracking should not be ignored when investigating the seismic fragility of the aging bridge systems. All these system fragility curves presented a more authentic estimation of the seismic vulnerability of aging bridges, and they offered a more accurate basis for life-cycle cost analysis.

## Figures and Tables

**Figure 1 materials-15-05020-f001:**
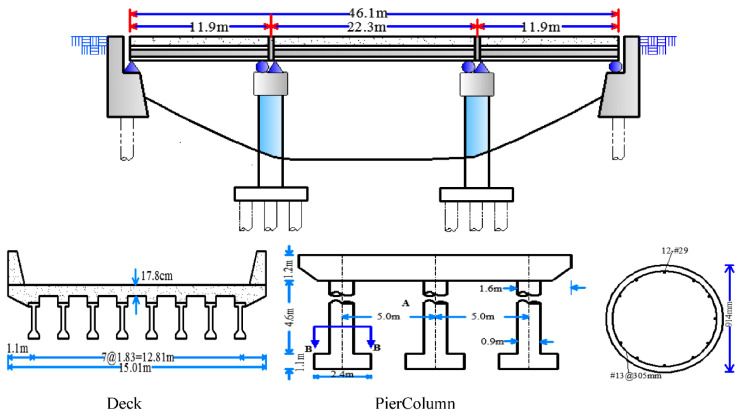
Case study multi-span continuous (MSC) concrete girder bridge.

**Figure 2 materials-15-05020-f002:**
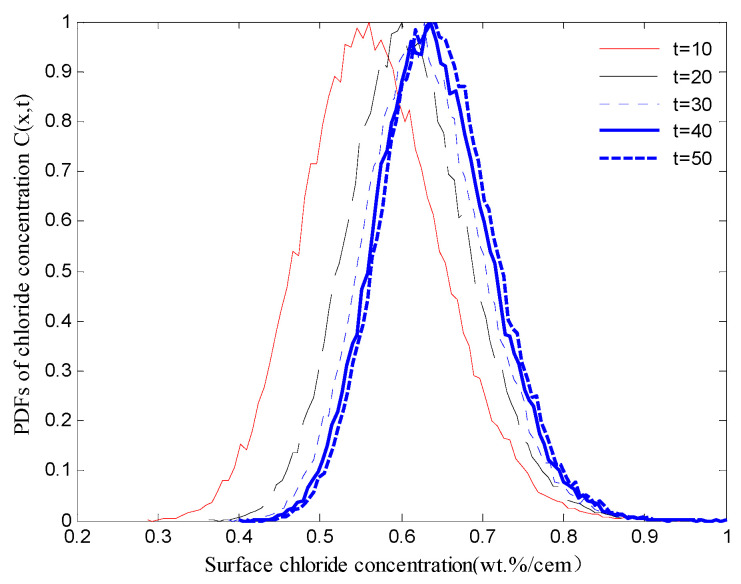
PDFs of the chloride concentration C(x,t).

**Figure 3 materials-15-05020-f003:**
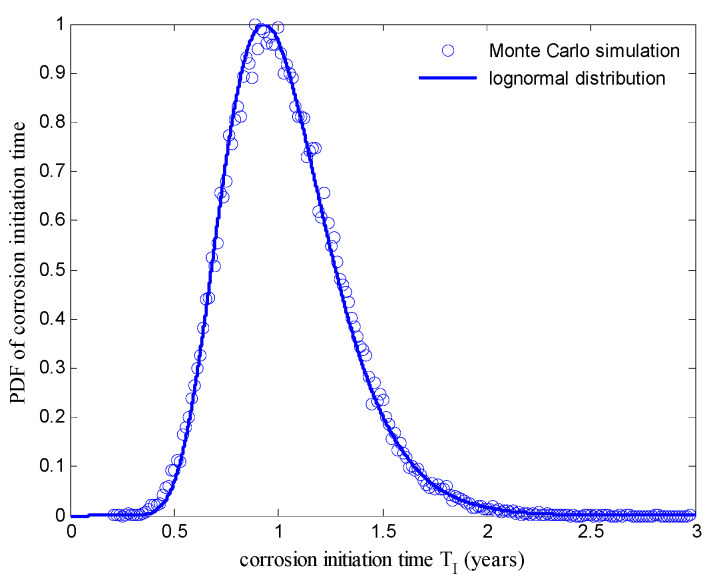
PDFs of the corrosion initiation time *T*_1_.

**Figure 4 materials-15-05020-f004:**
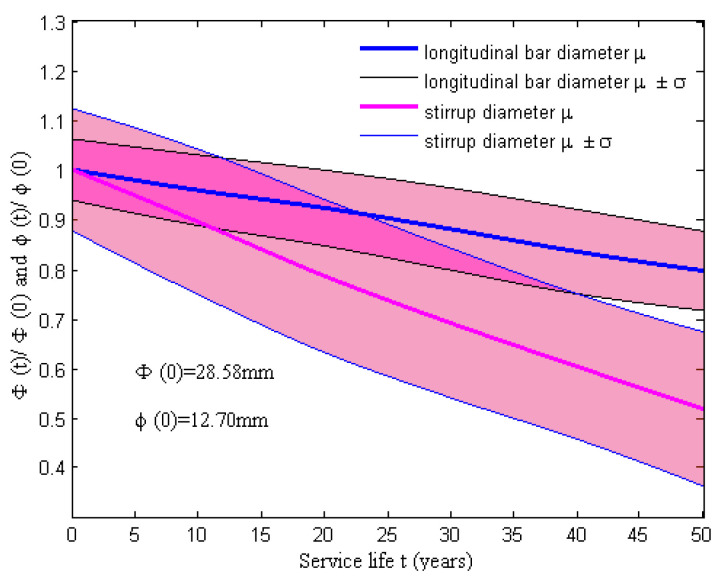
Time-evolving longitudinal bar and stirrup diameters for $\Phi \left(0\right)$ = 28.58 mm and $\phi \left(0\right)$ = 12.70 mm.

**Figure 5 materials-15-05020-f005:**
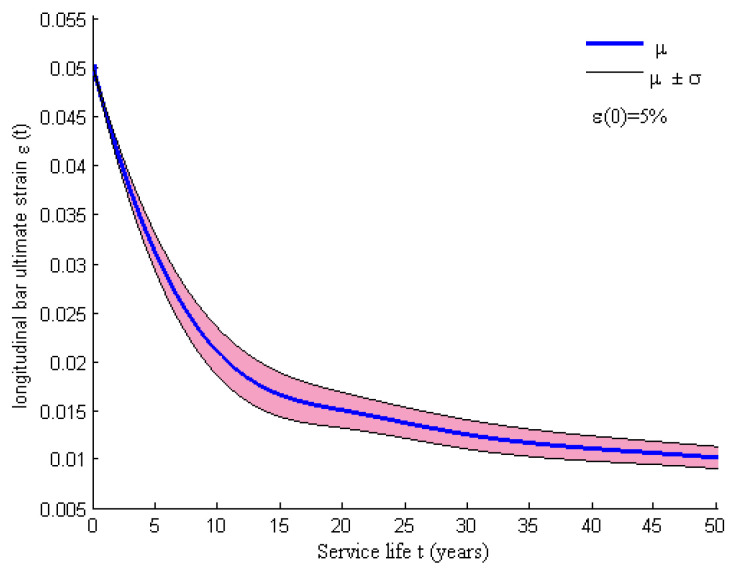
Ultimate strain of a longitudinal bar over time.

**Figure 6 materials-15-05020-f006:**
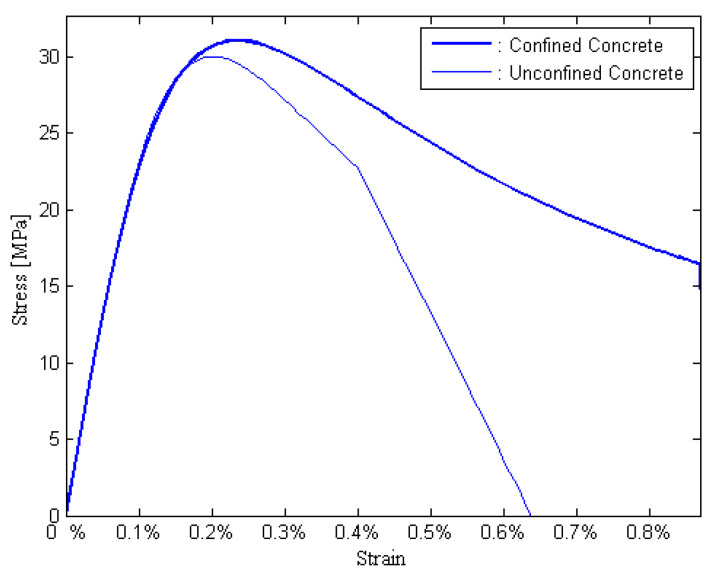
Unconfined and confined stress–strain law obtained with the BLG model.

**Figure 7 materials-15-05020-f007:**
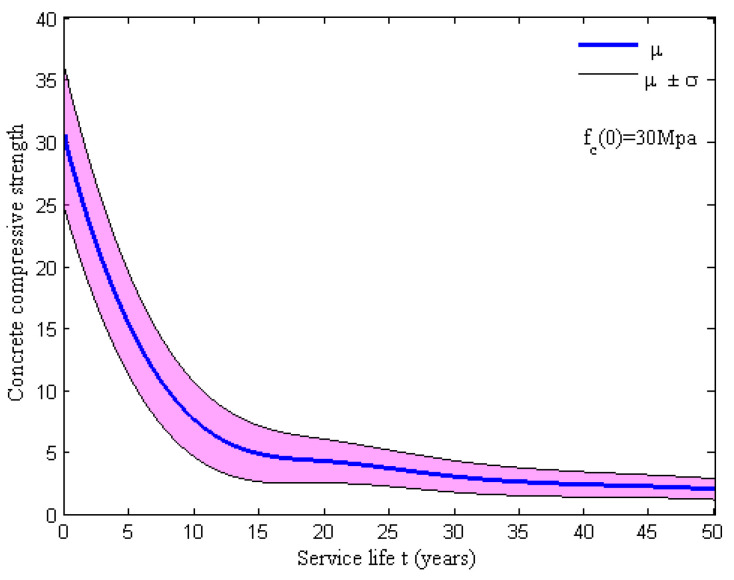
Time variance of the compression strength of unconfined concrete.

**Figure 8 materials-15-05020-f008:**
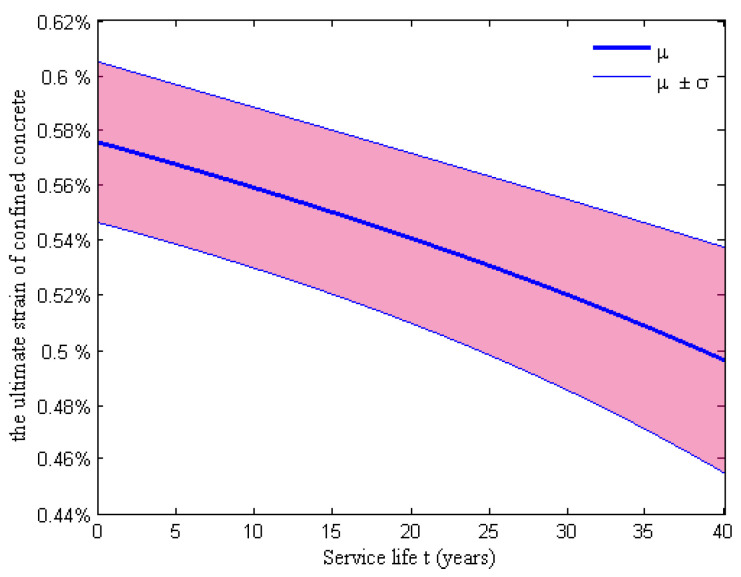
Time-dependence of the ultimate compressive strain of confined concrete.

**Figure 9 materials-15-05020-f009:**
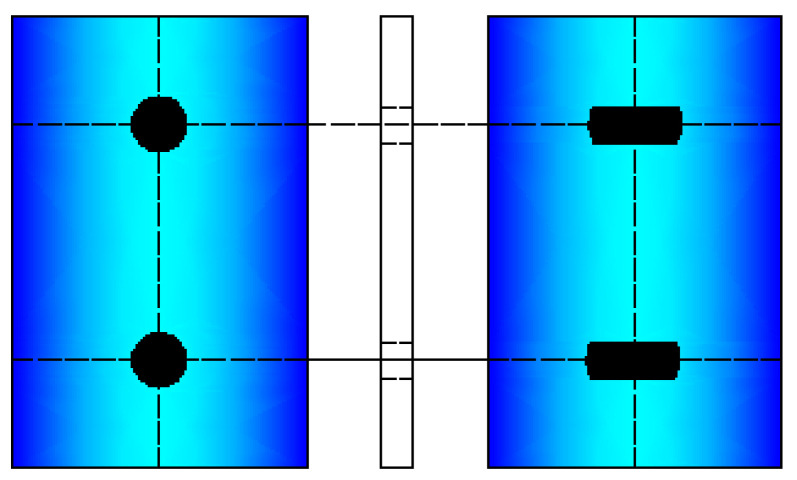
Fixed and expansion elastomeric bearing types, depending on the dimensions of the slot.

**Figure 10 materials-15-05020-f010:**
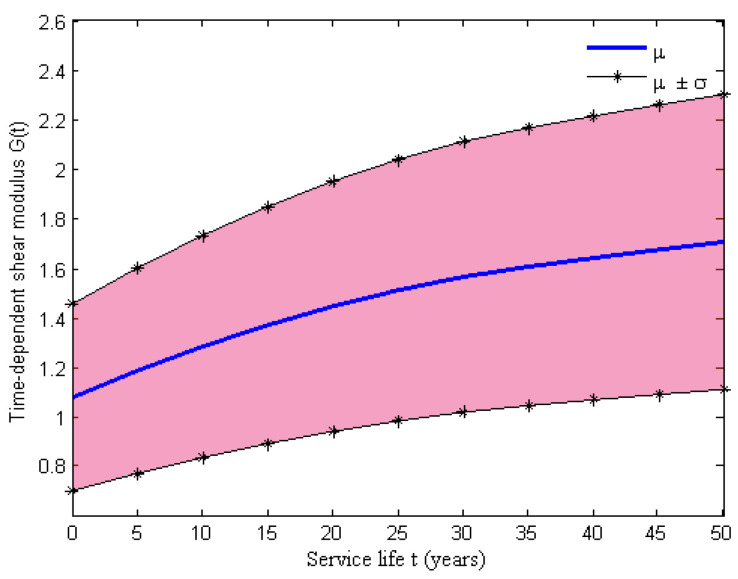
Time evolution of the shear modulus in a bearing NR pad.

**Figure 11 materials-15-05020-f011:**
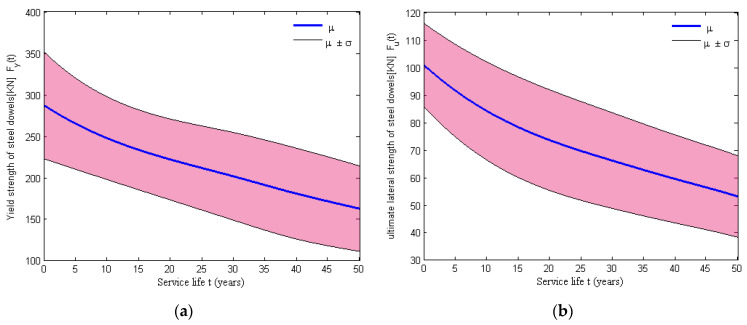
Time evolution of the (**a**) yield strength and (**b**) ultimate lateral strength for swedged steel dowels.

**Figure 12 materials-15-05020-f012:**
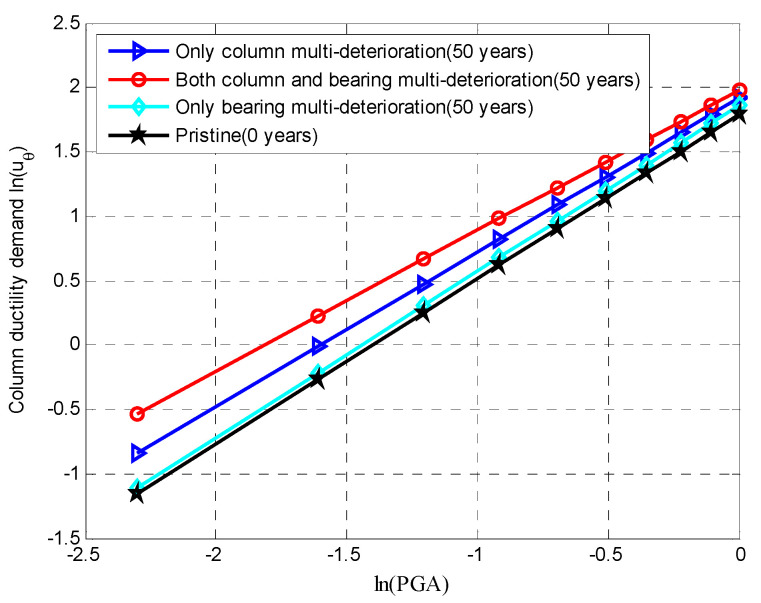
Mean value of the demand placed on an RC column through PSDM.

**Figure 13 materials-15-05020-f013:**
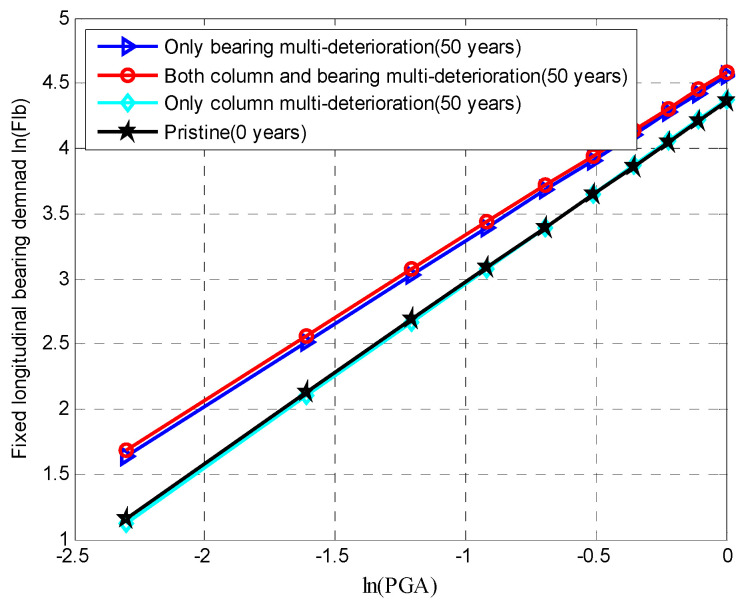
Mean values of the demand placed on a fixed bearing in the longitudinal direction using a PSDM.

**Figure 14 materials-15-05020-f014:**
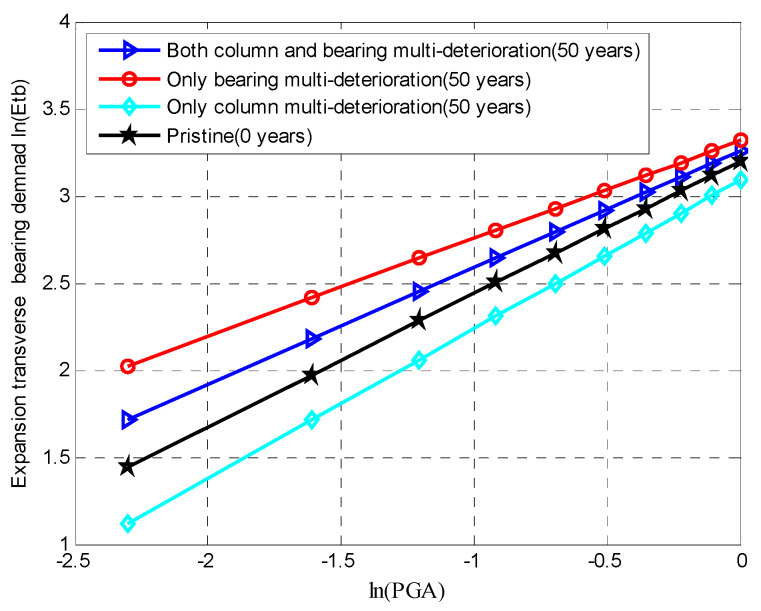
Mean values of the demand placed on an expansion bearing in the transverse direction using a PSDM.

**Figure 15 materials-15-05020-f015:**
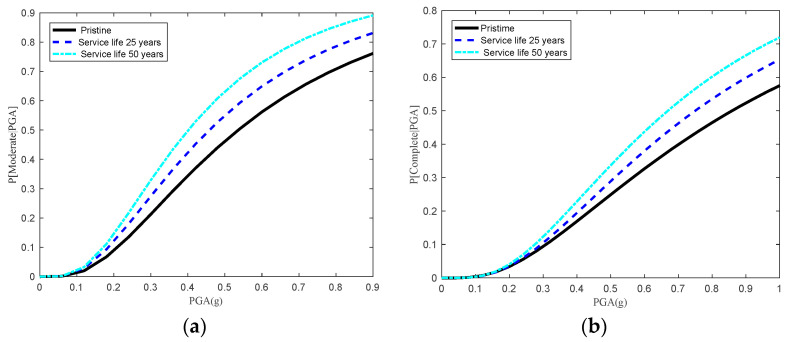
Time-dependent system seismic fragility curves for an MSC concrete girder bridge for (**a**) a moderate damage state and (**b**) a complete damage state under multi-deterioration of multiple components.

**Figure 16 materials-15-05020-f016:**
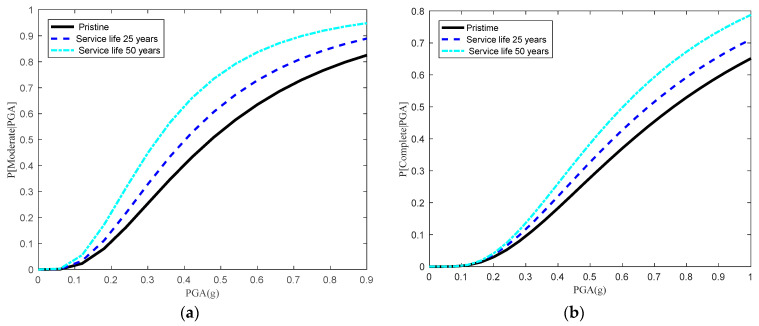
Time-dependent system seismic fragility curves for an MSC concrete girder bridge for (**a**) a moderate damage state and (**b**) a complete damage state subjected to prestress losses and cracking.

**Table 1 materials-15-05020-t001:** Equation list for Section 2.

Equation Number	Equation Expression
Equation (1)	$\frac{\partial C(x,t)}{\partial t}=\frac{\partial }{\partial x}D\frac{\partial C(x,t)}{\partial x}$
Equation (2)	$C(x,t)=C_{S}\left[1-erf\left(\frac{x}{2\sqrt{Dt}}\right)\right]$
Equation (3)	$T_{I}=\frac{x^{2}}{4D}{\left[erf^{-1}\left(\frac{C_{S}-C_{C}}{C_{S}}\right)\right]}^{-2}$
Equation (4)	$R\left(t\right)=\int_{T_{I}}^{t}r_{corr}dt$
Equation (5)	$\Phi \left(t\right)=\Phi \left(0\right)-2R\left(t\right)$
Equation (6)	$\phi (t)=\phi \left(0\right)-2R\left(t\right)$
Equation (7)	$\Phi \left(t\right)=\Phi_{i}-R\left(t\right)$
Equation (8)	$\phi \left(t\right)=\phi_{i}-R\left(t\right)$
Equation (9)	$\eta =2R\left(t\right)/\Phi \left(0\right)$ ; η∈[0,1]
Equation (10)	$\lambda =2R\left(t\right)/\phi \left(0\right)$ ; λ∈[0,1]
Equation (11)	$A(t)=\left\{\begin{array}{l} \left[1-\eta_{S}\left(t\right)/2\right]A_{S}\left(0\right),\;\;t\leq T_{I}\\ \left[1-\eta_{S}\left(t\right)\right]A_{S}\left(0\right),\;\;T_{I}\leq t\leq T_{I}+\Phi_{i}/r_{corr}\\ 0,\;\;t\geq T_{I}+\Phi_{i}/r_{corr} \end{array}\right.$
Equation (12)	$a(t)=\left\{\begin{array}{l} \left[1-\lambda_{k}\left(t\right)/2\right]a_{s}\left(0\right),\;\;t\leq T_{I}\\ \left[1-\lambda_{k}\left(t\right)\right]a_{s}\left(0\right),\;\;T_{I}\leq t\leq T_{I}+\phi_{i}/r_{corr}\\ 0,\;\;t\geq T_{I}+\phi_{i}/r_{corr} \end{array}\right.$
Equation (13)	$\eta_{S}\left(t\right)=\eta \left(2-\eta \right)$
Equation (14)	$\varepsilon_{su}\left(t\right)=\left\{\begin{array}{ll} \varepsilon_{su}\left(0\right) & 0\leq \eta_{S}\leq 0.016\\ 0.1521\eta_{S}{}^{-0.4583}\varepsilon_{su}(0{)}^{\;\;\;,} & 0.016<\eta_{S}\leq 1 \end{array}\right.$
Equation (15)	$f_{sy}\left(t\right)=\left(1-1.077\eta_{S}\left(t\right)\right)f_{sy}\left(0\right)$
Equation (16)	$f_{yh}\left(t\right)=\left(1-1.077\lambda_{k}\left(t\right)\right)f_{yh}\left(0\right)$
Equation (17)	$f_{pc}\left(t\right)=\frac{f_{pc}(0)}{1+K\varepsilon_{t}\left(t\right)/\varepsilon_{c}\left(0\right)}$
Equation (18)	$\varepsilon_{t}\left(t\right)=\frac{n_{bars}w}{r_{i}}$
Equation (19)	$w=k_{w}\left(\Delta A_{s}-\Delta A_{s0}\right)$
Equation (20)	$\eta_{S}\left(0\right)=1-{\left[1-\frac{\alpha }{\phi \left(0\right)}\left(7.53+9.32\frac{x}{\phi \left(0\right)}\right)10^{-3}\right]}^{2}$
Equation (21)	$\varepsilon_{cu}(t)=0.004+0.9\rho_{s}(t)\frac{f_{yh0}}{300}$

**Table 2 materials-15-05020-t002:** Probability distribution and the random parameters involved in the diffusion process.

Random Variable	Unit	Distribution Type	Mean	COV
Concrete cover, x	mm	Lognormal	40.00	0.20
Diffusion coefficient, D	cm^2^/year	Lognormal	1.29	0.10
$\mathrm{Surface}\;\mathrm{chloride}\;\mathrm{concentration},\;C_{s}$	wt%/cem	Lognormal	0.10	0.10
$\mathrm{Critical}\;\mathrm{chloride}\;\mathrm{concentration},\;C_{C}$	wt%/cem	Lognormal	0.040	0.10
$\mathrm{Corrosion}\;\mathrm{rate},\;r_{corr}$	mm/year	Lognormal	0.127	0.3

**Table 3 materials-15-05020-t003:** Probability distribution of random parameters for bridge modeling.

Uncertainty Parameter	Units	Distribution Type	Distribution Parameters	
			A ^1^	B ^1^
Concrete compressive strength	Mpa	Lognormal	30	5/30
Reinforcing steel yield strength	Mpa	Lognormal	300	30/300
Reinforcing steel diameter	mm	Lognormal	28.58	0.10
Stirrup yield strength	Mpa	Lognormal	235	30/235
Stirrup diameter	mm	Lognormal	12.70	0.10
Elastomeric bearing shear modulus	Mpa	Uniform	0.66	2.07
Steel dowel lateral strength	Mpa	Uniform	200.96	381.70
Dowel bar tension strength	Mpa	Uniform	522.09	845.00

^1^ Mean and standard deviation of the lognormal distribution, lower bound and upper bound of the uniform distribution.

**Table 4 materials-15-05020-t004:** Equation list for Section 3.

Equation Number	Equation Expression	Equation Number	Equation Expression
Equation (22)	$B_{s}/B_{0}=k_{s}\sqrt{t_{r}}+1$	Equation (32)	$\frac{G\left(t\right)}{G_{0}}=1+\tau \Delta G_{s}$
Equation (23)	$k_{s}=a_{1}\xi^{3}+a_{2}\xi^{2}+a_{3}\xi +a_{4}$	Equation (33)	$\tau =\left\{\begin{array}{l} {\left((m-h)/h\right)}^{2}\\ 0\\ {\left((m-(l-h))/h\right)}^{2} \end{array}\right.$ $\begin{array}{l} \left(0\leq m\leq h\right)\\ \left(h\leq m\leq l-h\right)\\ \left(l-h\leq m\leq l\right) \end{array}$
Equation (24)	$\begin{array}{l} B_{s}=G_{s}(\lambda_{1}^{2}+\lambda_{2}^{2}+\lambda_{3}^{2}-3)\\ B_{0}=G_{0}(\lambda_{1}^{2}+\lambda_{2}^{2}+\lambda_{3}^{2}-3) \end{array}$	Equation (34)	$G\left(t\right)=G_{0}\left(1+\tau \Delta G_{s}\right)$
Equation (25)	$G_{s}/G_{0}=B_{s}/B_{0}=k_{s}\sqrt{t_{r}}+1$	Equation (35)	$t_{r}/t=e^{\frac{E}{R}(\frac{1}{T_{r}}-\frac{1}{T})}$
Equation (26)	$\Delta G_{s}=(G_{s}-G_{0})/G_{0}=G_{s}/G_{0}-1$	Equation (36)	$\Delta G_{s}(t)=k_{s}\sqrt{t}$
Equation (27)	$h=\alpha \exp(\beta /T^{*})$	Equation (37)	$k\left(t\right)=G\left(t\right)A_{p}/S_{p}$
Equation (28)	$G(t)/G_{0}=B(t)/B_{0}=1+\Delta G_{s}\;\;\;\;\;\;\;\;\;\;\;\;\;\;\;\;(m=0\;\mathrm{or}\;l)$	Equation (38)	$F_{y}(t)=f_{y}A_{d}(t)/\sqrt{3}$
Equation (29)	$G(t)/G_{0}$$=1\;(h_{0}\leq m\leq l-h$)	Equation (39)	$F_{u}(t)=f_{u}A_{d}(t)/\sqrt{3}$
Equation (30)	dG(t)/dm=0 (m=h or l−h)		
Equation (31)	$\frac{G\left(t\right)}{G_{0}}=v_{1}m^{2}+v_{2}m+v_{3}$		

**Table 5 materials-15-05020-t005:** Equation list for Section 4 and Section 5.

Equation Number	Equation Expression	Equation Number	Equation Expression
Equation (40)	$u_{\theta }=\frac{\theta_{m}}{\theta_{y}}$	Equation (49)	$T_{p}=f_{p}A_{p}\left(\eta \right)$
Equation (41)	$P_{f}\left(t\right)=P\left[D\left(t\right)\geq C\left(t\right)|IM\right]$	Equation (50)	$\rho =4\pi \left(R_{0}^{2}-R_{\rho }^{2}\right)/A_{p}$
Equation (42)	$\ln\left[{\hat{D}}_{i}\left(t\right)\right]=\ln\left[a_{i}\left(t\right)\right]{+b}_{i}(t)\ln(IM)\;$	Equation (51)	$w_{c}=\frac{4\pi (R_{t}-R_{0})}{A^{*}+B^{*}}-H^{*}$ $\left\{\begin{array}{l} A^{*}=(1-\nu_{c}){\left(R_{0}/R_{c}\right)}^{\sqrt{a}}\\ B^{*}=(1+\nu_{c}){\left(R_{c}/R_{0}\right)}^{\sqrt{a}}\\ H^{*}=2\pi R_{c}f_{t}/E_{c} \end{array}\right.$
Equation (43)	$\beta_{Dm,i}\left(t\right)=\sqrt{\frac{\sum {\left(\ln\left(d_{m,i}\left(t\right)\right)-\ln\left(a_{i}\left(t\right)IM_{m}^{b_{i}\left(t\right)}\right)\right)}^{2}}{N-2}}$		
Equation (44)	$\begin{array}{l} P\left[D_{i}\left(t\right)\geq d_{i}\left(t\right)|IM\right]\\ =\Phi \left(\frac{\ln\left(IM\right)-\lambda_{i}\left(t\right)}{\xi_{i}\left(t\right)}\right) \end{array}$		
Equation (45)	$P_{i}\left(DS|IM\right)=\Phi \left(\frac{\ln\left(IM\right)-\gamma_{i}\left(t\right)}{\zeta_{i}\left(t\right)}\right)$		
Equation (46)	$\gamma_{i}\left(t\right)=\frac{\ln\overset{\wedge }{C_{i}\left(t\right)}-\ln a\left(t\right)}{b\left(t\right)}$		
Equation (47)	$\zeta_{i}\left(t\right)=\frac{\sqrt{\beta_{Dm,i|}^{2}\left(t\right)+\beta_{C,i}^{2}\left(t\right)}}{b\left(t\right)}$		
Equation (48)	$P_{sys}\left(DS|IM\right)=\Phi \left(\frac{\ln\left(IM\right)-\ln\left[\gamma_{sys}\left(t\right)\right]}{\zeta_{sys}\left(t\right)}\right)$		

**Table 6 materials-15-05020-t006:** Capacity limit state for different bridge components for an MSC concrete girder bridge.

Bridge	Slight		Moderate		Extensive		Complete	
Component	Med.	Disp.	Med.	Disp.	Med.	Disp.	Med.	Disp.
RC columns	1.29	0.59	2.10	0.51	3.52	0.64	5.24	0.65
Fixed bearing—longitudinal	28.9	0.60	104.2	0.55	136.1	0.59	186.6	0.65
Fixed bearing—transverse	28.8	0.79	90.9	0.68	142.2	0.73	195.0	0.66
Expansion bearing—longitudinal	28.9	0.60	104.2	0.55	136.1	0.59	186.6	0.65
Expansion bearing—transverse	28.8	0.79	90.9	0.68	142.2	0.73	195.0	0.66

**Table 7 materials-15-05020-t007:** Applicability of different approaches.

Different Approaches	Applicability
Theoretical modeling of the corrosion process combined with finite element modeling	Deals with multi-deterioration mechanisms among multiple components (e.g., RC columns, bearing systems); Deals with the deterioration process of prestress loss and cracking; Develops the time-dependent overall seismic fragility of aging bridge systems; Updates the system fragility assessment of a bridge with real-time monitoring data.
Empirical experiment and trial-and-error method [51,52,53]	It is very suitable for a single component’s corrosion process; It is time-consuming for time-dependent overall fragility of aging bridge systems and the earthquake excitation test; It is very expensive and very unrealistic to simultaneously obtain a series of multi-deterioration mechanisms, prestress loss and cracking among multiple components (e.g., RC columns, bearing systems) of aging bridge systems.

## Data Availability

No data were used to support this study.

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
