# Peer review of "Time-Dependent Seismic Fragility of Typical Concrete Girder Bridges under Chloride-Induced Corrosion"

_materials, 2022, doi:10.3390/ma15145020_

Round 1
Reviewer 1 Report
The reviewer thanks the authors for the work done. The manuscript cannot be selected as a "research article". In fact, the authors listed a number of reference models based on literature review to analyze the conditions of concrete bridge-girders under corrosion. In the last part, instead, the authors suggested that fragility curves, according to finite element modeling, can uniquely represent such analyses. Nevertheless, I recommend the publication of the manuscript as a "review article" in the “Materials, MDPI” whether following major modifications will be taken into account within the text:
1) The current state of knowledge relating to the topic is not covered and clearly presented, and the authors’ contribution is not emphasized. In this regard, the authors should make their effort to address these issues, by adding additional comments on the state of the art and the proposed aspects of their "review article".
2) Introduction. Corrosion of steel reinforcements and prestressing strands of concrete bridges provokes significant prestress losses and cracking which, in turn, induce excessive deflections during their service life. Please, refer to these issues, and cite the following references:
- Damage detection in a post tensioned concrete beam – Experimental investigation, Eng. Struct. 128 (2016) 15–25.
- Experimental–theoretical investigation of the short-term vibration response of uncracked prestressed concrete members under long-age conditions. Structures, 2022, 35, pp. 260–273.
3) Section 1. The Finite Element (FE) model is meaningless. No data has been elaborated. Please, delete the FE model from Section 1.
4) Sections 2-5. Too many formulas have been inserted. Please, try to summarize the contents of the expressions using graphs and tables within the text.
5) Reference models that treat prestress losses and cracking in concrete bridge-girders should be illustrated and, subsequently, be proposed as part of the fragility curves.
6) Title, abstract and conclusions must then be revised based on the aforementioned comments.
7) I suggest to the authors to edit all the text of the article with the help of a native English speaker. Grammar, punctuation, spelling, verb usage, sentence structure, conciseness, readability and writing style must also be improved. One of the editing services is reported at: https://www.mdpi.com/authors/english.
Author Response
Responds to the reviewers’ comments:
Reviewer #1:
Major comments:
Reviewer #1: The reviewer thanks the authors for the work done. The manuscript cannot be selected as a "research article". In fact, the authors listed a number of reference models based on literature review to analyze the conditions of concrete bridge-girders under corrosion. In the last part, instead, the authors suggested that fragility curves, according to finite element modeling, can uniquely represent such analyses. Nevertheless, I recommend the publication of the manuscript as a "review article" in the “Materials, MDPI” whether following major modifications will be taken into account within the text:
1) The current state of knowledge relating to the topic is not covered and clearly presented, and the authors’ contribution is not emphasized. In this regard, the authors should make their effort to address these issues, by adding additional comments on the state of the art and the proposed aspects of their "review article".
Response: Our authors have added additional comments on the state of the art and have provided the aspects on the review. The revision parts have been marked in red. Please see lines 31 to 100.
Meanwhile, the authors’ contributions have been elaborated. The revision parts have been marked in red and have been highlighted in yellow in lines 101 to 137.
2) Introduction. Corrosion of steel reinforcements and prestressing strands of concrete bridges provokes significant prestress losses and cracking which, in turn, induce excessive deflections during their service life. Please, refer to these issues, and cite the following references:
- Damage detection in a post tensioned concrete beam – Experimental investigation, Eng. Struct. 128 (2016) 15–25.
- Experimental–theoretical investigation of the short-term vibration response of uncracked prestressed concrete members under long-age conditions. Structures, 2022, 35, pp. 260–273.
Response: Our authors have referred to these issues and have cited these references. The revision parts have been marked in red and have been highlighted in yellow. See lines 71 to 74 and section 5.4.
3) Section 1. The Finite Element (FE) model is meaningless. No data has been elaborated. Please, delete the FE model from Section 1.
Response: The FE model from Section 1 has been deleted.
4) Sections 2-5. Too many formulas have been inserted. Please, try to summarize the contents of the expressions using graphs and tables within the text.
Response: From sections 2-5, our authors have used tables to summarize the contents of the expressions. See Tables 3-6.
5) Reference models that treat prestress losses and cracking in concrete bridge-girders should be illustrated and, subsequently, be proposed as part of the fragility curves.
Response: Our authors have added section 5.4 to illustrate Reference models that treat prestress losses and cracking in concrete bridge-girders and have proposed these models as part of the fragility curves. The revision parts have been marked in red and section 5.4.
6) Title, abstract and conclusions must then be revised based on the aforementioned comments.
Response: Our authors have revised title, abstract and conclusions based on the aforementioned comments. The revision parts have been marked in red. See lines 1-2, lines 12-29, and lines 603-632. Meanwhile, all of references have been updated and have been marked in red. Seen section References.
7) I suggest to the authors to edit all the text of the article with the help of a native English speaker. Grammar, punctuation, spelling, verb usage, sentence structure, conciseness, readability and writing style must also be improved. One of the editing services is reported at: https://www.mdpi.com/authors/english.
Response: Based on all of the reviewer’s comments, our authors have edited all the text of the article with the help of a native English speaker. Grammar, punctuation, spelling, verb usage, sentence structure, conciseness, readability and writing style can be improved through “Certificate of Language Editing”, which can be found in the attachment. Considering the convenient of aspects, our authors have chosen the translation institution, which has been long-term cooperation with our institution. In order to further convenient for reimbursing publishing charges, we are requested to choose the translation institution, assigned by our institution. Thanks for understanding.
Special thanks to you for your good comments.

Reviewer 2 Report
1. The concrete girder bridge model is used in the paper, mention reasons for using this model.
2. Finite Element method is used to solve the problem, is this the only approach?
3. How does this approach compare with any other approach in this field and why it is positive? Use a chart or table etc highlight this point.
Author Response
Comments and Suggestions for Authors
- The concrete girder bridge model is used in the paper, mention reasons for using this model.
Response: Some existing literatures [1-5, 23] have indicated that the corrosion of the reinforcement steel in RC columns and the deterioration of the bridge bearings could be found in MSC steel girder bridges and MSSS concrete girder bridges in Central and Southeastern United States, which are seismically vulnerable regions. Meanwhile, according to China’s Ministry of Transportation, the aging and deterioration phenomenon of MSC concrete girder bridges are commonly found in seismic zones such as northern China because the deicing salts are extensively used in highway bridges [24]. In addition, the typical MSC concrete girder bridge identified by Nielson (2005) is introuduced in this study, because of the similar bridge configuration in China [24]. Furthermore, nearly 67.8% of all bridges in Central and Southeastern United States and approximately 74% in China are constituted by this type of the bridges [24]. The MSC concrete girder bridge is used for the case study, since an overwhelming majority of 74% bridges are found in seismic zones and these bridges are seismically vulnerable due to inadequate detailing of components. Previous studies on classes of bridges by Nielson [31] have indicated that the vulnerability of multiple components of this type of bridges are prior to considering aging and deterioration. This phenomenon can be attributed to inadequately seat widths, bolted elastomeric pad bearings and the insufficient transverse reinforcement which inhibits the shear resistance and ductile capacity in RC column.
Consequently, in order to verify the proposed methodology for developing the time-dependent fragility curves and offer insights on the effects of multi-deterioration mechanism of multiple related components on seismic response and vulnerability, a sample MSC concrete girder bridge is considered as a case study in this paper.
The mention reasons have been highlighted in yellow. See lines 89-95 and lines 142-152.
[1] Choe D-E, Gardoni P, Rosowsky D, Haukaas T. Seismic fragility estimates for reinforced concrete bridges subject to corrosion. Structural Safety. 2009; 31(4):275-283.
[2] Choe D-E, Gardoni P, Rosowsky D, Haukaas T. Probabilistic capacity models and seismic fragility estimates for RC columns subject to corrosion. Reliability Engineering & System Safety. 2008;93(3):383-393.
[3] Li J, Gong J, Wang L. Seismic behavior of corrosion-damaged reinforced concrete columns strengthened using combined carbon fiber-reinforced polymer and steel jacket. Construction and Building Materials. 2009; 23(7):2653-2663.
[4] Alipour A, Shafei B, Shinozuka M. Performance evaluation of deteriorating highway bridges located in high seismic areas. Journal of Bridge Engineering. 2011; 16(5):597-611.
[5] Liu, M., Cheng, X., Li, X., Yue, P., & Li, J. (2016). Corrosion behavior and durability of low-alloy steel rebars in marine environment. Journal of Materials Engineering and Performance, 2016, 25(11), 4967-4979.
[23] Ghosh J, Padgett JE. Impact of multiple component deterioration and exposure conditions on seismic vulnerability of concrete bridges. Earthquake and Structures. 2012; 3(5):649-673.
[24] M Liu, Effect of uniform corrosion on mechanical behavior of E690 high-strength steel lattice corrugated panel in marine environment: A finite element analysis. Materials Research Express, 2021, 8(6), 066510.
[31] Nielson BG. Analytical fragility curves for highway bridges in moderate seismic zones [Ph.D.]. Ann Arbor: Georgia Institute of Technology; 2005.
- Finite Element method is used to solve the problem, is this the only approach?
Response: Actually, before Finite element methods for modelling the degradation process of the MSC girder bridges, we should use theoretical models of corrosion degradation to analyze the degradation process of RC components (e.g. columns and bearings). After we obtain a series of theoretical process of multi-deterioration among multiple components, then the code program for the theoretical process can be embed into the finite element modeling. However, such the empirical experiment method and trial-and-error method is very expensive and very unrealistic for obtaining a series of multi-deterioration process of the bridge systems including multiple RC components subject to earthquake excitations because the time-dependent seismic fragility of the aging bridge systems is required in this research. Thus, if one would like to efficiently obtain the entire time-variant deterioration process of aging bridge systems, the finite elements modeling combining with the theoretical analysis is the first choice in developing the time-dependent seismic fragility curves of aging bridge systems. For the other methods such as the empirical experiment and trial-and-error cannot be suggested in this research.
- How does this approach compare with any other approach in this field and why it is positive? Use a chart or table etc highlight this point
Response: It should be stressed that other time-variant approaches such as the empirical experiment and trial-and-error [51-53] for corrosion process of aging bridge systems are very expensive and very unrealistic to obtain a series of the multi-deterioration mechanisms among multiple components of aging bridge systems. Moreover, the empirical experiment and trial-and-error is very consuming for dealing with the time-dependent system fragility of such complex aging bridges, suffering from the earthquake excitation test. Therefore, when we would like to efficiently develop the overall time-variant deterioration process of aging bridge systems, the finite elements modeling combing with the theoretical modeling of corrosion process is the best choice for performing the time-dependent overall seismic fragility curves of aging bridge systems. The applicability of different approaches can be summarized in Table 8.
Table 8 Applicability of different approaches
Different approaches |
Applicability |
Theoretical modeling of corrosion process combining with Finite element modeling |
1. Deal with multi-deterioration mechanisms among multiple components (e.g. RC columns, bearing systems); 2. Deal with the deterioration process of prestress loss and cracking; 3. Develop the time-dependent overall seismic fragility of aging brides systems; 4. Update the system fragility assessment of bridge with the real-time monitoring data. |
Empirical experiment and trial-and-error method [51-53] |
1. It is very suitable for single component’s corrosion process; 2. It is time-consuming for time-dependent overall fragility of aging bridge systems and the earthquake excitation test; 3. It is very expensive and very unrealistic to simultaneously obtain a series of the multi-deterioration mechanisms, prestress loss, and cracking among multiple components (e.g. RC columns, bearing systems) of aging bridge systems. |
[51] Nie, J., Braverman, J., Hofmayer, C., Choun, Y. S., Kim, M. K., & Choi, I. K. Fragility analysis methodology for degraded structures and passive components in nuclear power plants I llustrated using a condensate storage tank (No. KAERI/TR--4068/2010). Korea Atomic Energy Research Institute, 2010.
[52] Ayazian, R., Abdolhosseini, M., Firouzi, A., & Li, C. Q. Reliability-based optimization of external wrapping of CFRP on reinforced concrete columns considering decayed diffusion. Engineering Failure Analysis, 2021, 128, 105592.
[53] Sun, H., Burton, H. V., & Huang, H. Machine learning applications for building structural design and performance assessment: State-of-the-art review. Journal of Building Engineering, 2021, 33, 101816.
The revision parts have been highlighted in yellow. See lines 601-610 and Table 8.
Special thanks to you for your good comments.

Round 2
Reviewer 1 Report
The authors have adequately addressed my review comments. Thus, I recommend the publication of the manuscript as a "REVIEW ARTICLE" in the “Materials, MDPI”.